# Drought Resistance and Recovery of Kentucky Bluegrass (*Poa pratensis* L.) Cultivars under Different Nitrogen Fertilisation Rates

Cristina Pornaro * , Michele Dal Maso and Stefano Macolino

Department of Agronomy, Food, Natural Resources, Animals, and Environment, University of Padova, 35020 Padova, Italy; michele.dalmaso@studenti.unipd.it (M.D.M.); stefano.macolino@unipd.it (S.M.)
* Correspondence: cristina.pornaro@unipd.it

**Abstract:** Kentucky bluegrass (*Poa pratensis* L.) is one of the most popular cool-season turfgrass species. However, little is known about the effects of N supply on its resistance to drought stress. The objective of this study was to assess the effects of acute drought followed by a recovery period on four Kentucky bluegrass cultivars ('Barduke', 'Brooklawn', 'NuBlue Plus' and 'Marauder') and one tall fescue ('Rhambler SRP') under two different nitrogen fertilisation rates (100 or 200 kg N ha$^{-1}$ yr$^{-1}$). The study was conducted over two years in a rain-out structure to control water input in spring and summer. The cultivars were subjected to a drought stress phase (absence of irrigation) followed by a recovery phase (weekly irrigation with an intake of 80% of ET). The green cover percentage, visual quality, NDVI, and soil moisture were measured weekly. We found that Kentucky bluegrass maintains sufficient turf quality for 2 weeks without irrigation. During the first year of the experiment, slight differences were observed among the Kentucky bluegrass cultivars, with 'Marauder' showing a 15% of green turf cover less than 'Brooklawn' after 6 weeks of acute drought, while in the second year, 'NuBlue Plus' displayed higher green turf cover and NDVI than the other cultivars. Nitrogen treatment had limited influence on the performances of the cultivars, 'Marauder' being the only one benefitting from the higher rate of applications.

**Keywords:** recovery from drought stress; visual quality; green turf cover; NDVI; soil moisture

## 1. Introduction

The Kentucky bluegrass (*Poa pratensis* L.) is one of the most popular cool-season turfgrass species for amenity and sport uses and is suitable for many environments with different climate. This dark green grass has a medium-fine leaf texture, high shoot density and persistence [1]. It has good wear tolerance, a high potential recovery rate due to its very vigorous rhizomes [2,3], and good heat resistance [4]. It is also considered to be a drought-tolerant species, although there is high variability among cultivars [5] mainly due to variations in apomictic reproduction rates [6]. The morphological and agronomic characteristics of this species mean it can be successfully combined with other species. In fact, Kentucky bluegrass is often mixed with tall fescue (*Schedonorus arundinaceus* (Schreb.) Dumort., syn. *Festuca arundinacea* Schreb.), perennial ryegrass (*Lolium perenne* L.), and fine fescue (*Festuca rubra* L.) [2].

It is widely recognised that climatic changes, particularly in recent years, have led to variations in temperature and in the distribution and intensity of rainfall [7]. Moreover, water availability is increasingly limited, and some authors have reported the need to reduce irrigation, especially in crops that are considered secondary, such as turfgrasses [5]. Drought has been predicted to become an important climatic stress factor in many regions [8], and it has been estimated that in some regions, such as the Mediterranean area, rainfall will decrease by up to 20% in the near future [7]. Kentucky bluegrass is an attractive turfgrass, but to maintain a desirable aesthetic appearance and maximise

seasonal greenness, a supplementary water supply in the form of irrigation is necessary [9]. Huang [10] classified the maximum evapotranspiration rate of Kentucky bluegrass as very high ($>10$ mm d$^{-1}$), although there are significant differences among cultivars.

Drought stress severely influences all agronomical, anatomical, and physiological attributes of Kentucky bluegrass. Under conditions of soil water deficiency, the number of leaves and shoots, the dry matter of roots and shoots, and the total root length decrease [11] and all these traits are useful to determine the drought resistance [12]. Drought tolerance generally requires a deep root system, which is important to fully utilise available soil water resources, thereby reducing the need for supplementary irrigation [5]. The combined effects of drought and heat stress are associated with damage to the cell membranes of grass species [13], which can reduce the leaf water content leading to a severe decline in turf quality under field conditions [13,14]. In the Mediterranean regions of Europe, the most critical period is summer when the combination of drought and heat stress markedly limit the performance of cool-season species [3].

Water and nitrogen (N) are the main factors affecting plant growth and quality. Nitrogen is a significant constituent for proper growth and development of plants and increases and enhances the yield and its quality by playing a vital role in biochemical and physiological functions of plants [15]. Nitrogen is a significant constituent of plants and influences plant metabolism under water stress conditions. Fertilisation has been reported to affect the drought stress resistance of Kentucky bluegrass [15], although it is important to apply N at the correct rate in order to not only improve drought tolerance, but also enhance the cellular antioxidant detoxification and osmoprotection functions [16,17]. The N requirements of Kentucky bluegrass range from 0.20 to 0.40 kg 100 m$^{-2}$ every 15–30 days of growth [2]. The frequency of N application plays a critical role in alleviating drought stress, since optimal nutrition levels sustain metabolic activities under reduced tissue water potential [18]. Under drought stress conditions, plant roots are unable to obtain optimal amounts of N from the soil, and this has a negative effect on plant growth and metabolism [19].

The use of drought-resistant species and cultivars is the primary strategy for reducing turfgrass water consumption. As water resources are also becoming increasingly limited in the Mediterranean countries of Europe, the selection of low water-demand varieties is crucial for establishing sustainable turfgrasses in these areas. The aim of this study was to investigate the performances of four Kentucky bluegrass cultivars under drought conditions and their recovery capacities. The four cultivars were compared with 'Rhambler SRP' tall fescue, which is known for its high drought avoidance due to a deep, efficient root system [4,20]. All the cultivars were fertilised with N at two different rates to test their performances under different rates of N availability. This study hypothesized high soil N availability could positively affect drought tolerance and that cultivar response to N application could vary greatly.

## 2. Materials and Methods

### 2.1. Experiment Description

The study was conducted at the Experimental Agricultural Farm of the University of Padua located in Legnaro, North-Eastern Italy (45°20′ N, 11°57′ E; elevation 8 m). The soil at the site is a coarse-silty, mixed, mesic Oxyaquic Eutrudept [21] containing 13% clay, 40% silt, and 47% sand, with a pH of 8.7, 2.5% organic matter, a C-to-N ratio of 10.5, an Olsen-P content of 6 mg kg$^{-1}$, and an exchangeable K content of 171 mg kg$^{-1}$. The climate of the area is classified as sub-humid, with a mean annual temperature of 12.6 °C and an annual rainfall of 830 mm falling mostly from April to November [22].

The experiment was carried out from September 2018 to August 2019 and was repeated from September 2019 to September 2020 in a rain-out structure (16 m × 8 m) to prevent rain from reaching the test plots. An ET gauge meter (Spectrum Technologies, Inc., Plainfield, IL) was installed in the structure. A portable weather station (WatchDog 2700 Station, Spectrum Technologies Inc., Plainfield, IL, USA) was also installed in the rain-out shelter to record hourly ground temperatures (°C) at a depth of 1.5 m.

In September 2018 and September 2019, four Kentucky bluegrass cultivars ('Barduke', 'Brooklawn', 'NuBlue Plus' and 'Marauder') were seeded at a rate of 25 g m$^2$, and the tall fescue cultivar Rhambler SRP was seeded at a rate of 40 g m$^2$ as the control. The experimental design was spilt-plot with three replications. Each plot (1.20 m × 3.80 m) was split into two sub-plots (1.20 m × 1.90 m) to test the two different N rates: low (100 kg ha$^{-1}$ yr$^{-1}$) and optimal (200 kg ha$^{-1}$ yr$^{-1}$), according to Croce et al. [2]. The annual nitrogen fertilisation programme consisted of four applications: December (20%), March (30%), May (20%), and September (30%). On this basis, in the low-fertilised sub-plots, 20 kg N ha$^{-1}$ was applied in mid-December, 30 kg N ha$^{-1}$ in mid-March, and 20 kg N ha$^{-1}$ in mid-May, while in the optimally fertilised sub-plots, 40, 60, and 40 kg N ha$^{-1}$ were applied in mid-December, mid-March, and mid-May, respectively. Fertiliser was not applied in September as the experiments were completed in August in 2019, and by early September in 2020. Nitrogen was applied manually with a granular urea fertiliser (46%).

Before seeding, the plots were fertilised with 50 kg N ha$^{-1}$, 150 kg P$_2$O$_5$ ha$^{-1}$ and 150 kg K$_2$O ha$^{-1}$. As they were becoming established, the turfgrasses were irrigated daily with 2 to 3 mm of water by means of an overhead sprinkler irrigation system. When full cover was reached, the plants were irrigated weekly at 100% evapotranspiration (ET) until drought stress was imposed. The plots were mowed to a height of 43 mm once a week during the growing season with a rotary mower (Honda HRX 476 SX E, Honda Motor Co. Ltd., Tokyo, Japan) and clippings were removed.

The cultivars were assessed in two phases: drought and recovery. The drought phase started on 24th June 2019 and 10th June 2020 and continued until the best performing cultivar reached 60% green turf cover, which was on 29th July 2019 and 28th July 2020. The percentage of green turf cover was measured by digital image analysis, as described by Richardson et al. [23]. The recovery phase started on 29th July 2019 and 28th July 2020, when weekly irrigation was resumed at 80% of ET as recorded by the gauge meter. The second phase was considered to have terminated when the best performing cultivar reached 80% green turf cover (29th August 2019 and 9th September 2020). Prior to both the drought and the recovery phases, the soil water content was restored to field capacity.

During the two phases (drought and recovery), the following parameters were measured weekly: soil moisture (Field Scout TDR 300 Soil Moisture Meter, Spectrum Technologies Inc., Plainfield, IL, USA), the normalised difference vegetation index (NDVI, GreenSeeker Handheld Crop Sensor, Trimble Navigation Unlimited, Sunnyvale, CA, USA), and the percentage of green turf cover (digital image analysis, [23]). Turf quality and colour were visually rated on a scale of 1 to 9 [24,25]. Average weekly temperatures during the experimental periods of each year (2019 and 2020) are reported in Figure 1.

## 2.2. Statistical Analysis

An analysis of variance was performed using a linear mixed-effect model to test the effects of year, cultivar, sampling date, N rate, and their interactions on the parameters measured (green turf cover, NDVI, soil moisture and visual turf quality). Data for each year (2019 and 2020) were analysed separately, as the length of the drought and recovery phases differed in the two years, so the sampling dates did not coincide. Green turf cover and NDVI data were squared transformed to ensure normality and homoscedasticity of the residuals; then, back transformed to obtain the final results. A least significant difference (LSD) test with Bonferroni correction at a probability of 0.05 was used to identify significant differences among means. To exclude spatial correlation due to the edge effect of the rain-out structure, the longitude and latitude of each plot were included as covariates (X and Y positions) in the model. Statistical differences were determined by likelihood ratio tests of the models with and without inclusion of position as a covariate. Position was not found to be significant for any of the parameters tested based on Akaike's Information Criterion (AIC). All statistical analyses were performed in R version 3.4.0 [26] and additionally the "nlme" package for fitting mixed models, and "multcomp" for post hoc comparisons.

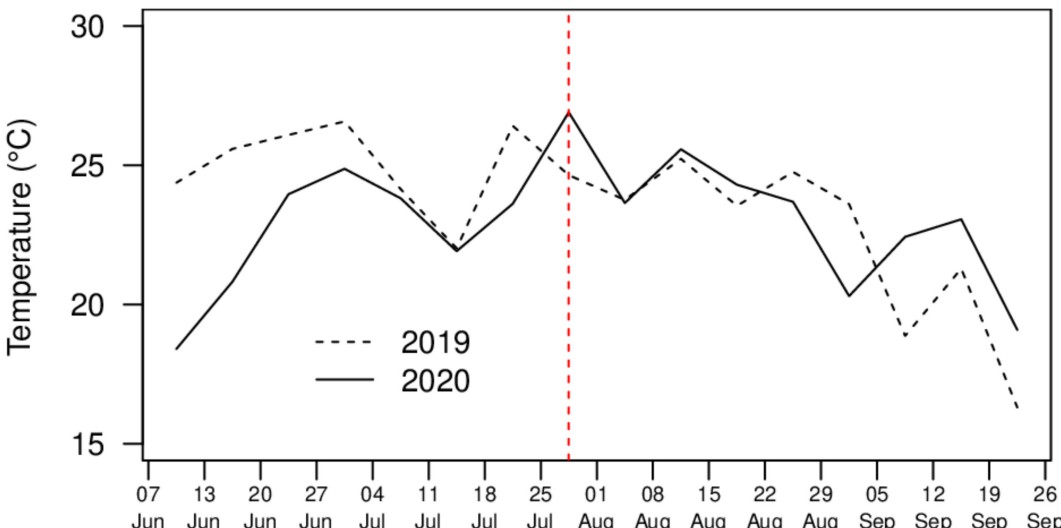

**Figure 1.** Weekly mean air temperatures in a rain-out structure in Legnaro (Padua, North-Eastern Italy) during the experimental periods of 2019 (June–August) and 2020 (June–September). The red vertical line indicates when the date irrigation recovery began (29th July 2019, 28th July 2020).

## 3. Results

The three-way interaction between the cultivar, sampling date and N rate was not significant in either year (Table 1). The interaction between the cultivar and sampling date was significant for all the parameters tested. The interaction between the cultivar and N rate was significant for NDVI and soil moisture in 2019, and for green turf cover, NDVI and visual quality in 2020.

### 3.1. Green Turf Cover

In 2019, a significant reduction in green turf cover was observed from the second week for all cultivars except 'NuBlue Plus' and 'Rhambler SRP', which showed a significant decrease in green cover from the third week (Figure 2a, Table S1). Five weeks after the beginning of the drought phase, 'Marauder' showed a significantly lower green cover than the others cultivars. However, due to the high reduction in green cover of 'Barduke' and 'NuBlue Plus' during the last week of the drought phase, in the sixth week the green cover of 'Marauder' (45%) was only lower than 'Rhambler SRP' (77%) and 'Brooklawn' (58%). During the first week of the recovery phase, all Kentucky bluegrass cultivars reached the same level of green cover as 'Rhambler SRP' except 'Marauder', which reached a level similar to the other cultivars only five weeks into the irrigation recovery phase. Green cover values were stable for all cultivars from one to five weeks after irrigation recovery, but even after six weeks of irrigation recovery they did not reach values similar to those at the beginning of the experiment.

In 2020, no significant decreases and no significant differences in green turf cover were observed among the cultivars during the first three weeks of the drought phase (Figure 2b, Table S2). From the fifth week until the end of the drought phase, the percentages of green cover of all the Kentucky bluegrass cultivars were lower than 'Rhambler SRP', and 'Marauder' was lower than 'NuBlue Plus'. A slight improvement was observed from the first week of irrigation recovery, although after six weeks none of the cultivars had reached the same green cover as at the beginning of the experiment. 'Marauder' had lower values than 'NuBlue Plus' for the first four weeks of the recovery phase, and all the Kentucky bluegrass cultivars had lower values than 'Rhambler SRP' for the whole recovery phase, except for 'NuBlue Plus' and 'Brooklawn' three weeks after irrigation recovery.

**Table 1.** Results of the analysis of variance testing the effects of the cultivar and sampling date and their interaction on the percentage of green turf cover, the normalised difference vegetation index (NDVI), soil moisture, visual quality (scale 1–9) and colour (scale 1–9).

| Source | 2019 | | | | 2020 | | | |
|---|---|---|---|---|---|---|---|---|
| | Green Turf Cover | NDVI | Soil Moisture | Visual Quality | Green Turf Cover | NDVI | Soil Moisture | Visual Quality |
| Cultivar (cv) | *** | *** | *** | *** | *** | *** | *** | *** |
| N rate (N) | n.s. | n.s. | n.s. | n.s. | n.s. | n.s. | n.s. | n.s. |
| Sampling date (da) | *** | *** | *** | *** | *** | *** | *** | *** |
| cv × N | n.s. | * | ** | n.s. | ** | *** | n.s. | *** |
| cv × da | *** | ** | * | *** | *** | *** | * | *** |
| N × da | n.s. | n.s. | n.s. | n.s. | n.s. | n.s. | n.s. | n.s. |
| cv × N × da | n.s. | n.s. | n.s. | n.s. | n.s. | n.s. | n.s. | n.s. |

* Significant F test at the 0.05 level of probability. ** Significant F test at the 0.01 level of probability. *** Significant F test at the 0.001 level of probability. n.s.: non-significant at the 0.05 level of probability.

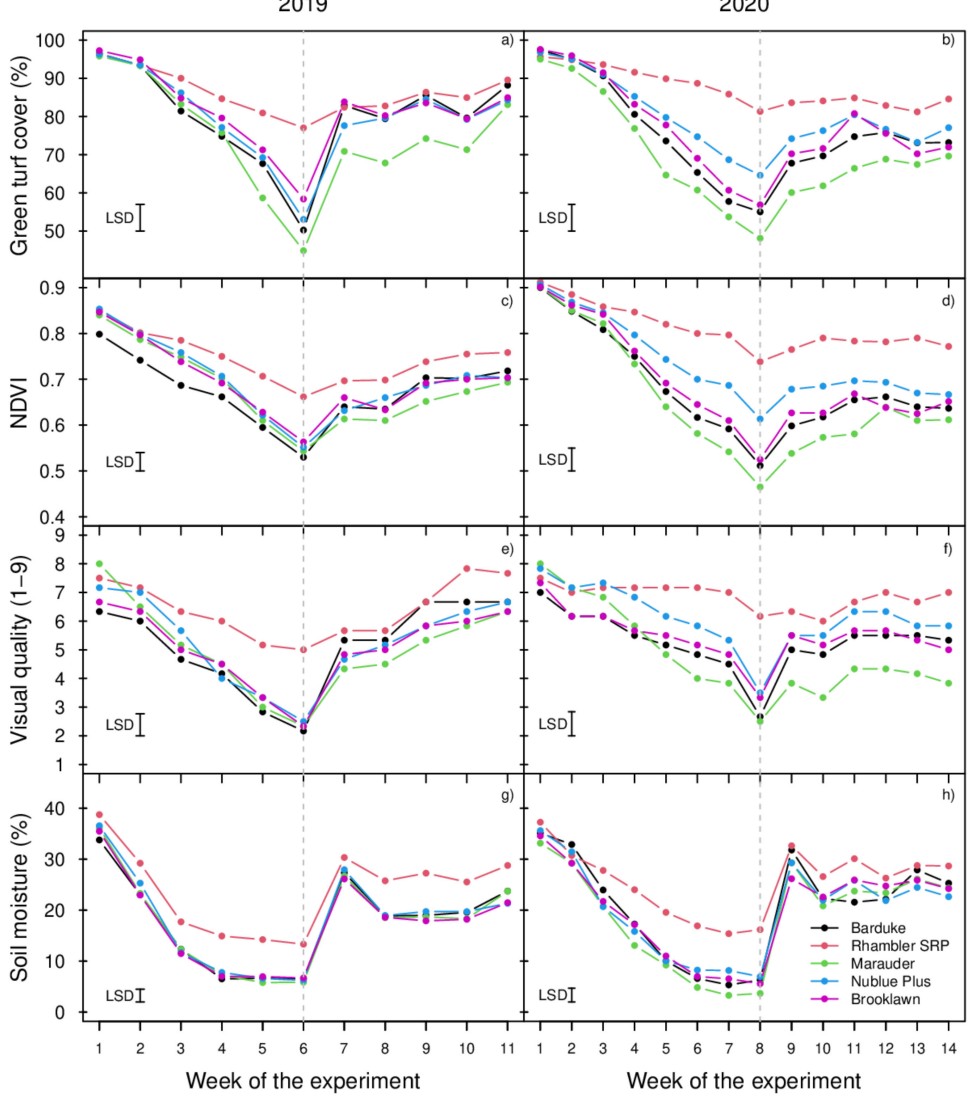

**Figure 2.** The effects of the sampling date on green turf cover (**a,b**), NDVI (**c,d**), visual quality (**e,f**), and soil moisture (**g,h**) in 2019 (on the left) and 2020 (on the right) of the four Kentucky bluegrass cultivars ('Barduke', 'Marauder', 'NuBlue Plus' and 'Brooklawn') and the tall fescue cultivar 'Rhambler SRP'. In all the graphs, the grey dashed line separates the drought phase on the left from the recovery phase on the right.

The significant interaction between the cultivar and N rate found in 2020, highlighted a different response of cultivars green turf cover as affected by N rates. Differences among cultivars between N rates were found for 'Barduke' only, which displayed lower green turf cover in the plots fertilised at the higher N rate (Figure 3), with values similar to both N rates of 'Marauder'. 'Rhambler SRP' displayed the highest values.

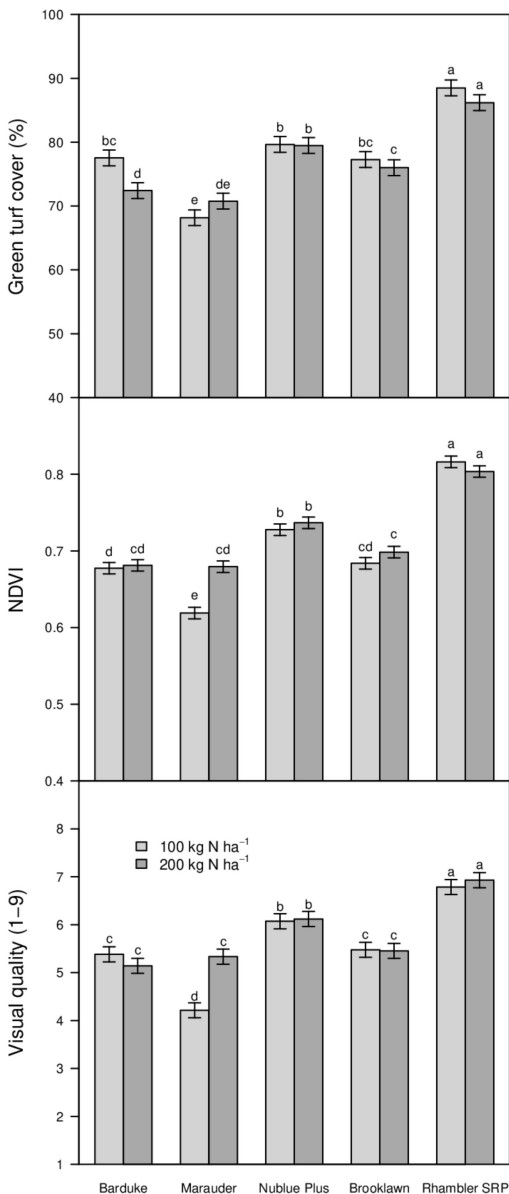

**Figure 3.** Green turf cover, NDVI and visual quality of four Kentucky bluegrass cultivars ('Barduke', 'Marauder', 'NuBlue Plus', 'Brooklawn') and a tall fescue cultivar ('Rhambler SRP') at two N fertilisation rates (100 and 200 kg ha$^{-1}$ yr$^{-1}$) in 2020 (June–September 2020). Mean values with the same letters are not significantly different based on the LSD test at a probability of 0.05. Bars are standard error.

### 3.2. Normalised Difference Vegetation Index

The normalised difference vegetation index of 'Barduke' was lower than all the other cultivars during the first three weeks of the drought phase in 2019 (Figure 2c, Table S1), while the other Kentucky bluegrass cultivars did not differ significantly from 'Rhambler SRP', with the exception of 'Brooklawn' in the third week. From the third week, 'Rhambler SRP' displayed higher NDVI values than all the other cultivars. During the recovery phase, all Kentucky bluegrass cultivars had lower values than 'Rhambler SRP', and differences

among the cultivars emerged only in the second and third week of irrigation recovery, with 'Marauder' having the lowest value.

The NDVI of the drought phase of 2020 followed a similar trend to green turf cover (Figure 2d, Table S2). At the end of the drought phase, the Kentucky bluegrass cultivar with the highest index was 'NuBlue Plus'. Recovery was difficult for all the cultivars, including 'Rhambler SRP', and there were differences among the Kentucky bluegrass cultivars in the first three weeks after irrigation recovery. The NDVI never reached the levels observed at the beginning of the experiment, and the Kentucky bluegrass cultivars never exceeded the value of 0.7.

The interaction between the cultivar and N rate was significant for both years. The effect of the N rate within cultivar was limited to 'Marauder' and 'NuBlue Plus' fertilised at the higher N rate in 2019 (Figure 4), and to 'Marauder' fertilized at the higher N rate in 2020 (Figure 3). In 2019, 'Rhambler SRP' displayed higher values than all Kentucky bluegrass cultivars under the higher N rate, while 'Rhambler SRP' fertilized with the lower N rate showed similar values than all Kentucky bluegrass cultivars fertilized with higher N rates, with the exception of 'Barduke'. In 2020, the highest values were observed for both N rates of 'Rhambler SRP', followed by 'NuBlue Plus', with lowest values for 'Marauder' fertilized with the lower N rate.

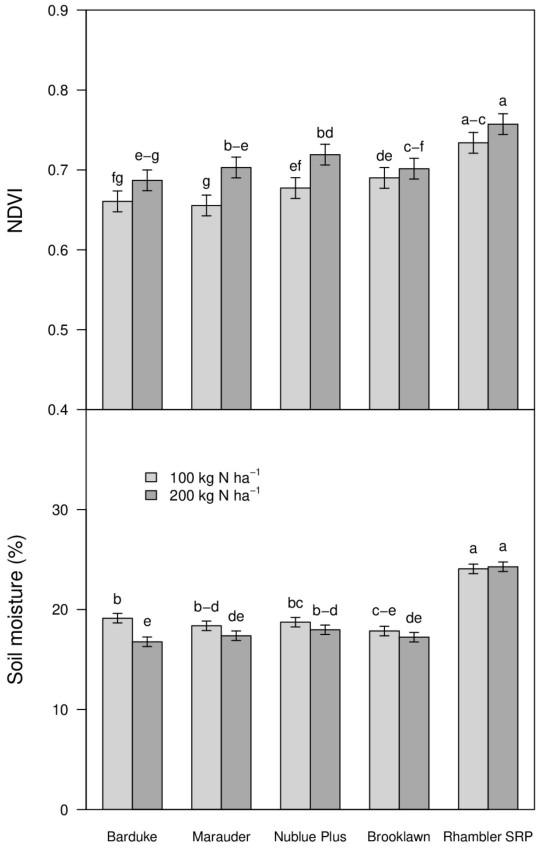

**Figure 4.** NDVI and soil moisture of four Kentucky bluegrass cultivars ('Barduke', 'Marauder', 'NuBlue Plus' and 'Brooklawn') and a tall fescue cultivar ('Rhambler SRP') at two N fertilisation rates (100 and 200 kg ha$^{-1}$ yr$^{-1}$) in 2019 (June–August). Means values with the same letters are not significantly different based on the LSD test at a probability of 0.05. Bars are standard error.

### 3.3. Visual Quality

During the drought phase of 2019, there were differences in visual quality among the various Kentucky bluegrass cultivars in the first three weeks (Figure 2e, Table S1). 'Marauder' displayed higher values than 'Barduke' and 'Brooklawn' in the first week, but in the second and third week 'NuBlue Plus' performed better than 'Barduke' and no worse

than 'Rhambler SRP'. Four weeks after the beginning of the drought phase, all cultivars displayed lower values than 'Rhambler SRP' and this gap remained until the first week of the irrigation recovery phase when 'Marauder' was the only cultivar to have a lower value than 'Rhambler SRP'. From the fourth week of the recovery phase, the visual quality levels of all the cultivars were similar and lower than 'Rhambler SRP'.

There were no significant decreases in visual quality in the first three weeks of the experiment during 2020 (Figure 2f, Table S2). 'Rhambler SRP' exhibited no significant changes during the whole experimental period. Among the Kentucky bluegrass cultivars, only 'Barduke' and Brooklawn' exhibited lower values than 'NuBlue Plus' in the third and fourth weeks, but from the fifth week 'Marauder' had lower values than 'NuBlue Plus'. At the end of the drought phase there were no differences among the Kentucky bluegrass cultivars. 'Marauder' did not significantly improve in visual quality during the recovery phase. 'NuBlue Plus' and 'Brooklawn' reached values similar to 'Rhambler SRP' one week after irrigation recovery, but from then on there were no further increases and the values at the end of the experiment were lower than six.

In 2020, the effect of the N fertilisation rate was observed for 'Marauder' only, which displayed the lowest visual quality in the plots fertilised at the lower N rate (Figure 3), while plot of 'Marauder' fertilized with the higher N rate reached values of both the N rates of 'Barduke' and 'Brooklawn'. 'NuBlue Plus' displayed an average rate of six for both N rates, with values higher than all others Kentucky bluegrass cultivars, but lower than 'Rhambler SRP'.

### 3.4. Soil Moisture

No differences in soil moisture among the Kentucky bluegrass cultivars were observed during 2019 (Figure 2g, Table S1). Higher values were recorded in the plots of 'Rhambler SRP' during the drought and the recovery phases, except in the first week of the study period and the first week after irrigation recovery.

During the drought phase of 2020, there was a greater decrease in soil moisture in the plots of Kentucky bluegrass cultivars (declining to around 10%) than in the plots of 'Rhambler SRP' (declining to 20%) (Figure 2h, Table S2). Tall fescue, with its greater green turf cover (Figure 2h, Table S1) and deeper root system [20] seemed to maintain water content better in the first 7 cm of soil. Differences among the Kentucky bluegrass cultivars were found only one week after irrigation recovery, with 'Barduke' having higher values than 'Brooklawn', while from the second week of the recovery phase 'NuBlue Plus' displayed lower values than 'Rhambler SRP'.

The interaction between the cultivar and N rate was significant in 2019 only. Soil moisture was lower in the plots of 'Barduke' fertilised at the higher N rates, while 'Rhambler SRP' displayed the highest values (Figure 4).

## 4. Discussions

### 4.1. Cultivar and Sampling Date Interaction

In 2020, the decrease in green cover was constant. The drought phase was longer than in 2019 and a decrease in green turf cover was recorded after the fifth week, probably due to the high temperatures of that week (Figure 1). The optimum temperature for shoot growth in cool-season grasses is 15–23 °C [27]. However, during the study period, mean temperatures often exceeded 22 °C (Figure 1), and the maximum temperature was over 26 °C in both experimental years (data not shown). In 2020, high temperatures were recorded in the first week of irrigation recovery (Figure 1) with negative consequences for the development of green cover. In 2020, both green cover and NDVI trends during the recovery phase indicated that all the Kentucky bluegrass cultivars, as well as 'Rhambler SRP' tall fescue, had difficulty recovering. The year 2020 appeared to be less critical than 2019 when small differences were observed among the Kentucky bluegrass cultivars, with 'Marauder' showing lower drought tolerance than 'Brooklawn'. In 2020, 'NuBlue Plus' displayed better tolerance than the other Kentucky bluegrass cultivars (Figure

2d), and during the recovery phase its green cover and visual quality were similar to 'Rhambler SRP'. Furthermore, in the first week of irrigation recovery in 2019, the turf responded quickly with a rapid increase in green cover and visual quality. These different behaviours of the cultivars in the two experimental years is not easy to explain. The higher temperatures during the drought phase in 2019 (Figure 1) may have significantly increased the negative impact of drought on all the cultivars tested, except 'Rhambler SRP' tall fescue. Indeed, the factors affecting drought-tolerance mechanisms are not well understood in this species, as remarked by Richardson et al. [5], who studied drought tolerance and its relationship with rooting capacity in 49 Kentucky bluegrass cultivars. They compared root weight from 30 to 60 cm soil depth and shoot weight growth under ideal growing conditions with the percentage of green cover of cultivars under acute drought stress without finding correlation between deep rooting and the ability to withstand long periods of water deficit. Contrasting results were found by Ebdon and Kopp [28] in a lysimeter study, who observed a relationship between deep rooting and reduced leaf firing in Kentucky bluegrass under drought stress, suggesting that deep rooting can play a role in drought tolerance of this species. Richardson et al. [5] suggested that other morphological or physiological parameters are involved in drought resistance of Kentucky bluegrass cultivars, including stomatal resistance, osmotic adjustment [29], reduced electrolyte leakage and increased photosynthetic efficiency [30]. Our results for turf quality reflect those of Liu et al. [31], who found turf quality values below six after 15 days of drought conditions. They subjected five Kentucky bluegrass cultivars to 21 days of drought stress at temperatures similar to our study and found at the end of the experiment that two cultivars still displayed values of four, while the others displayed values of two. These results suggest that Kentucky bluegrass maintains sufficient turf quality for 2 weeks without irrigation and is therefore less drought tolerant than perennial ryegrass subjected to the same drought conditions in the same environment [32]. In this last study, 11 cultivars were tested but no significant interaction between the cultivar and sampling date for turf quality was found, which shows that there is less variability in the response of perennial ryegrass to stress than in the response of Kentucky bluegrass, as Pornaro et al. [3] also observed for traffic stress. Although three of the four Kentucky bluegrass cultivars were able to quickly reach sufficient turf quality during the recovery phase, most turfgrass managers wish to maintain a good visual appearance during drought periods. The poor visual quality of all the cultivars in 2019 and most of them in 2020 during the drought phase suggests the need for supplementary irrigation starting about 2 weeks from the onset of drought conditions.

### 4.2. Cultivar and N Rate Interaction

Plant response to water scarcity is an intricate mechanism involving physiological characteristics [33]. When the plant water level falls below a critical point, stomata closure causes a reduction in the transpiration rate, which restricts water transportation to the plant [31]. Liu et al. [31] showed that an increase in available nutrition helps attenuate the negative effect of drought stress. N application could therefore play an important role in mitigating the negative effects of water stress. Saud et al. [16] reported that a suitable supply of nutrition alleviated drought stress damage in Kentucky bluegrass by sustaining metabolic activities. In contrast, in our study the N treatment had little effect on the performances of the cultivars, which suggests that the interaction between drought stress and N fertilisation needs further investigation. 'Marauder' was the only cultivar to benefit from a higher N fertilisation rate, while neither the higher nor the lower rates affected the performances of the other cultivars. Croce et al. [2] considered that N fertilisation at a rate of almost 240 kg ha$^{-1}$ year$^{-1}$ every 30 days for Kentucky bluegrass maintenance would probably be excessively costly.

## 5. Conclusions

The selection of low-water demand varieties is becoming ever more important. It is widely recognised in literature that there is high variability in the stress tolerances of different Kentucky bluegrass cultivars. In our study, we found that Kentucky bluegrass maintained sufficient turf quality for two weeks after the onset of acute drought and that differences among the cultivars were accentuated under higher temperatures. 'Marauder' showed less tolerance to drought stress than the other Kentucky bluegrass cultivars, while 'NuBlue Plus' displayed better tolerance than the others and a good capacity for recovery. However, performances were influenced by temperature, with high temperatures reducing drought stress tolerance in all the cultivars tested.

Differently from our hypothesis, the higher N fertilisation was not able to make up for drought stress, although, as expected, the response varied among cultivars. The worst performing cultivar was 'Marauder' which benefitted from the higher N fertilisation rate. It seems that increasing the N supply is beneficial only to those cultivars most severely affected by drought stress. However, the increase in turf quality due to a greater N supply is limited, and it does not justify the greater quantity of nitrogen required.

**Supplementary Materials:** The following are available online at https://www.mdpi.com/article/10.3390/agronomy11061128/s1, Table S1. The effects of sampling date on green turf cover, NDVI, visual quality, and soil moisture in 2019 of the four Kentucky bluegrass cultivars ('Barduke', 'Marauder', 'Nublue Plus' and 'Brooklawn') and the tall fescue cultivar 'Rhambler SRP'; Table S2. The effects of sampling date on green turf cover, NDVI, visual quality, and soil moisture in 2020 of the four Kentucky bluegrass cultivars ('Barduke', 'Marauder', 'Nublue Plus' and 'Brooklawn') and the tall fescue cultivar 'Rhambler SRP'.

**Author Contributions:** Conceptualization, S.M.; methodology, S.M.; software, C.P.; validation, C.P.; formal analysis, C.P.; investigation, M.D.M.; resources, S.M.; data curation, C.P.; writing—original draft preparation, C.P. and M.D.M.; writing—review and editing, C.P. and S.M.; visualization, C.P.; supervision, S.M.; project administration, S.M.; funding acquisition, S.M. All authors have read and agreed to the published version of the manuscript.

**Funding:** This research was funded by Padana Sementi Elette S.r.l.

**Institutional Review Board Statement:** Not applicable.

**Informed Consent Statement:** Not applicable.

**Conflicts of Interest:** The authors declare no conflict of interest.

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
