# Peer review of "Drought Resistance and Recovery of Kentucky Bluegrass (Poa pratensis L.) Cultivars under Different Nitrogen Fertilisation Rates"

_agronomy, doi:10.3390/agronomy11061128_

Round 1
Reviewer 1 Report
General comments
I have read the manuscript (agro-1223876). Entitle: Drought resistance and recovery of Kentucky bluegrass (Poa pratensis L.) cultivars under different nitrogen fertilization rates written by Cristina Pornaro et. al., for publication of agronomy. Author did the research on application of nitrogen (N) on the resistance of the different cultivars of the Kentucky bluegrass under the drought and recovery phase. Author found species specific response under the drought stress conditions. Author did the research well to include the imported component of the introduction concentration in the water stress conditions and which is obvious application potential and manuscript is valuable. However, I found some lacking in the most of the section such as introduction, result parts as well as discussion. Discussion section is shallow not well connect with the drought resistance mechanism and less connect to the introduction section. Author should also focus the effect of the drought stress in the plants such as alternation of morphology and physiology properties and also slightly also focus on the biochemical mechanism with the appropriate references. I request to author to use the latest references rather than the old reference that had published before 2000. Now I request to author for MAJOR revision of this manuscript. Furthermore, the I request to authors for revision according to the rules of the journal and correct the bibliography.
Major suggestions
1) Abstract and manuscript whole: Abstract should to be improved by minimize the descriptive sentences which is not suited in this section. Abstract should more logical, short, concise, and informative. Your abstract should reflect your study and major findings while shortly observed by readers. Please made the necessary corrections. Please see the Ln. 19-24, in these line author compares the different cultivars of turf grass between the years of their performance. However, author should represent the result more solid way, author should summaries the two years’ data and present the overall performance and give the resistance rank of those cultivars based on their overall until two years under the drought conditions. Based on this theme the whole manuscript should change the flow of writing.
2) Introduction and Discussion: These both sections lacking the appropriate information and relevant references and potential connection of these two components. Without appropriate literatures, questions or hypotheses in introduction section the entirely text will be unclear. So please addressed adequately with justifiable interpretation with sufficient literatures and text.
Others comments and suggestions
1) Line no. 17
“weekly irrigation at 80% ET” the recovery phase the amount of water or scheduling is not much clear please rephrase this.
2) Line no. 22-23
Please rephrase this is not the good flow, “Nitrogen treatment had limited influence 22 on the performances of the cultivars, ‘Marauder’ being the only one benefitting from the higher rate 23 of applications”.
3) Line no. 42-44
Author should to be present the negative effect of drought stress in broad perspective not only cover the turf grass because drought is the big figure of your study. Please write the text “drought reduced the morphological traits such as reduction of leaf size and vegetative growth, and physiological traits such as reduction of photosynthesis and stomatal conductance and alter the stem anatomical features” and cited this reference. DOI:10.1016/j.scienta.2018.11.021 Entitled: Impact of drought stress on photosynthesis responses, leaf water potential and stem sap flow of two cultivars….
4) Line no. 50-59
Author mention very well the different traits such as anatomical, morphological, biochemical biomass, shoot and root performance with the references but author should add one more sentences with the reference that the all those traits are very useful to determine the drought resistance please see the reference https://doi.org/10.1016/j.scitotenv.2021.146466 and added that the these all traits are the potential indicators for the drought resistance indicator (ref)”
5) Line no. 79
Please include the hypothesis of the research in the last section of the introduction
include the strong hypothesis of this research introduction because without appropriate hypotheses in introduction the entire introduction will be weaker. Please mention the text that such why you are doing this and what is the future insight of the research?
6) Line no. 132
While you indicate the meteorological data specially temperature please indicate the drought year such as 2019 or 2020 instead of Year 1 and Year 2. Accordingly, in the MS you should be thoroughly check and correct if mistake like that. You should represent the year.
7) Line no. 151
Please write the “statistical analysis” in separate in the independent subsection and more concise the analytic techniques the data. I saw you are also using the ANOVA but throughout the text of statistical analysis you are not mention to this.
8) Line no. 163
Please write the How your analysis the Table 1, SPSS? Or ANOVA by SAS? If you are using ANOVA, please give the detail information for the statistical text there. For e.g., which statistical software used for the figures? and which statistical package for e. g. version of SAS then mentions Version and other detail for example (SAS v. 9.4 (SAS Institute, Cary, NC, USA).
9) Line no. 183
As I said before please mention the fiscal year data in figure 2 instead of mention the year 1 and Year 2. Also in you figure 2 it is not sure that the 1,2,3,4…accordingly. It is confusing to the readers.
Please write the How your analysis the Table 1, SPSS? Or ANOVA by SAS? If you are using ANOVA, please give the detail information for the statistical text there. For e.g., which statistical software used for the figures?
10) Line no. 188
Author mention that the Result and Discussion together. However, according to author the text is not matching and randomized. I recommended to distinct the separate section like “Result” and “Discussion”.
In Scientific writing author write the both sections together should to be frequently follow the reference in in each important finding but I not see this in your Ms.
11) Line no. 289
Author should improve the figures, author should show the SE in each bar and use the letter or Asterisk (*) sign to shows the significant different. Please more focus the figures parts and analysis. It’s too shallow in your manuscript.
12) Line no. 315
Conclusion is comparatively weak. Author mostly repeat the same result part twice see the Ln 320-323. However, conclusion should be some more solid not the result repetition. I agree some result might have repeated but the way of presentation should be change. Please alter the expression as possible. Actually conclusions should be present the future insight of the research based on your current finding and strength of your results for the future research guideline of the related research.
13) Line no. 338
Reference: please double check the citations, its style and spell check and other grammatical errors. moreover, I request to authors for revision throughout the manuscript according to the journal rules.
Good Luck !
Author Response
General comments
I have read the manuscript (agro-1223876). Entitle: Drought resistance and recovery of Kentucky bluegrass (Poa pratensis L.) cultivars under different nitrogen fertilization rates written by Cristina Pornaro et. al., for publication of agronomy. Author did the research on application of nitrogen (N) on the resistance of the different cultivars of the Kentucky bluegrass under the drought and recovery phase. Author found species specific response under the drought stress conditions. Author did the research well to include the imported component of the introduction concentration in the water stress conditions and which is obvious application potential and manuscript is valuable. However, I found some lacking in the most of the section such as introduction, result parts as well as discussion. Discussion section is shallow not well connect with the drought resistance mechanism and less connect to the introduction section.
Discussion section has been improved and connected to introduction.
Author should also focus the effect of the drought stress in the plants such as alternation of morphology and physiology properties and also slightly also focus on the biochemical mechanism with the appropriate references.
Drought stress effects on morphology and physiology properties has been focused on the discussion section.
I request to author to use the latest references rather than the old reference that had published before 2000.
The references published before 2000 have been deleted, with the exception of two for which no more recent references have been found.
Now I request to author for MAJOR revision of this manuscript. Furthermore, the I request to authors for revision according to the rules of the journal and correct the bibliography.
The manuscript and the reference section have been formatted as request using the Microsoft Word template from the journal website.
Major suggestions
1) Abstract and manuscript whole: Abstract should to be improved by minimize the descriptive sentences which is not suited in this section. Abstract should more logical, short, concise, and informative. Your abstract should reflect your study and major findings while shortly observed by readers. Please made the necessary corrections.
The abstract has been improved in the result part to make it more informative.
Please see the Ln. 19-24, in these line author compares the different cultivars of turf grass between the years of their performance. However, author should represent the result more solid way, author should summaries the two years’ data and present the overall performance and give the resistance rank of those cultivars based on their overall until two years under the drought conditions. Based on this theme the whole manuscript should change the flow of writing.
Data have been analysed for separated years to avoid misunderstandings. The alterative way to present data was to build models with years included as random effect (as we did in the manuscript “Drought Stress Response of Turf-Type Perennial Ryegrass Genotypes in a Mediterranean Environment” – Agronomy 2020). However, in the present study, the overlapping data for the two years gives averaged values not representing the real. This is due to different lengths of drought and recovery periods in the two years: 6 weeks of drought and 5 weeks of recovery in the first year against 8 weeks of drought and 6 weeks of recovery in the second year.
2) Introduction and Discussion: These both sections lacking the appropriate information and relevant references and potential connection of these two components. Without appropriate literatures, questions or hypotheses in introduction section the entirely text will be unclear. So please addressed adequately with justifiable interpretation with sufficient literatures and text.
Some references about drought stress effects on morphology and physiology properties has been added in the discussion section according to your suggestions. If reviewers have any specific reference in mind that could improve the literature review, we would be happy to include it.
Others comments and suggestions
1) Line no. 17
“weekly irrigation at 80% ET” the recovery phase the amount of water or scheduling is not much clear please rephrase this.
Rephrased.
2) Line no. 22-23
Please rephrase this is not the good flow, “Nitrogen treatment had limited influence 22 on the performances of the cultivars, ‘Marauder’ being the only one benefitting from the higher rate 23 of applications”.
Rephrased.
3) Line no. 42-44
Author should to be present the negative effect of drought stress in broad perspective not only cover the turf grass because drought is the big figure of your study. Please write the text “drought reduced the morphological traits such as reduction of leaf size and vegetative growth, and physiological traits such as reduction of photosynthesis and stomatal conductance and alter the stem anatomical features” and cited this reference. DOI:10.1016/j.scienta.2018.11.021 Entitled: Impact of drought stress on photosynthesis responses, leaf water potential and stem sap flow of two cultivars….
Using references about trees might be out of context.
4) Line no. 50-59
Author mention very well the different traits such as anatomical, morphological, biochemical biomass, shoot and root performance with the references but author should add one more sentences with the reference that the all those traits are very useful to determine the drought resistance please see the reference https://doi.org/10.1016/j.scitotenv.2021.146466 and added that the these all traits are the potential indicators for the drought resistance indicator (ref)”
Reference has been added.
5) Line no. 79
Please include the hypothesis of the research in the last section of the introduction
include the strong hypothesis of this research introduction because without appropriate hypotheses in introduction the entire introduction will be weaker. Please mention the text that such why you are doing this and what is the future insight of the research?
Hypothesis has been added.
6) Line no. 132
While you indicate the meteorological data specially temperature please indicate the drought year such as 2019 or 2020 instead of Year 1 and Year 2. Accordingly, in the MS you should be thoroughly check and correct if mistake like that. You should represent the year.
“Year 1” and “Year 2” have been changed with “2019” and “2020” respectively.
7) Line no. 151
Please write the “statistical analysis” in separate in the independent subsection and more concise the analytic techniques the data. I saw you are also using the ANOVA but throughout the text of statistical analysis you are not mention to this.
Subsections have been added in the materials and methods accordingly.
It is redundant to specify in the results or discussion sections that the used analysis was ANOVA as we used only one analysis.
8) Line no. 163
Please write the How your analysis the Table 1, SPSS? Or ANOVA by SAS? If you are using ANOVA, please give the detail information for the statistical text there. For e.g., which statistical software used for the figures? and which statistical package for e. g. version of SAS then mentions Version and other detail for example (SAS v. 9.4 (SAS Institute, Cary, NC, USA).
The statistical software used for analysis is specified in the materials and methods as well as the statistical analysis performed. The software used for figures is no informative according to the aims of the manuscript.
9) Line no. 183
As I said before please mention the fiscal year data in figure 2 instead of mention the year 1 and Year 2. Also in you figure 2 it is not sure that the 1,2,3,4…accordingly. It is confusing to the readers.
“Year 1” and “Year 2” have been changed with “2019” and “2020” respectively. Title of X axis has been added.
Please write the How your analysis the Table 1, SPSS? Or ANOVA by SAS? If you are using ANOVA, please give the detail information for the statistical text there. For e.g., which statistical software used for the figures?
See comment above.
10) Line no. 188
Author mention that the Result and Discussion together. However, according to author the text is not matching and randomized. I recommended to distinct the separate section like “Result” and “Discussion”.
In Scientific writing author write the both sections together should to be frequently follow the reference in in each important finding but I not see this in your Ms.
The “Results and discussions” section has been separated in “results” and “discussions”.
11) Line no. 289
Author should improve the figures, author should show the SE in each bar and use the letter or Asterisk (*) sign to shows the significant different. Please more focus the figures parts and analysis. It’s too shallow in your manuscript.
SE and letters have been added, and the figure has been better described.
12) Line no. 315
Conclusion is comparatively weak. Author mostly repeat the same result part twice see the Ln 320-323. However, conclusion should be some more solid not the result repetition. I agree some result might have repeated but the way of presentation should be change. Please alter the expression as possible. Actually conclusions should be present the future insight of the research based on your current finding and strength of your results for the future research guideline of the related research.
Conclusions have been improved.
13) Line no. 338
Reference: please double check the citations, its style and spell check and other grammatical errors. moreover, I request to authors for revision throughout the manuscript according to the journal rules.
The manuscript and the reference section have been formatted as request using the Microsoft Word template from the journal website.
Good Luck !
Reviewer 2 Report
The authors have presented a manuscript, which evaluated Poa pratensis cultivars under different nitrogen fertilization. Cultivars were exposed to drought stress phase followed by a recovery phase. Visual parameters, NDVI, digital images parameters and soil moisture were evaluated. The manuscript presents interesting results concerning the tolerance of Poa turf to drought, but they are some point which need to improve. Following, I have included some comments aimed to enhance the paper:
- In Material and methods line 94-95 four Kentucky bluegrass cultivars (‘Barduke’, ‘Brooklawn’, ‘Nublue Plus’, ‘Maurauder’) were seeded. Cultivars have been cited briefly. Explain why you have chosen these varieties and give some characteristics of the chosen varieties (for example, which are more resistant to water stress, the most productive…) differences between them.
- The Table 1. Authors have presented levels of significance of cultivar and N application, but they have explained only the differences in the interaction. Explain all data presented in the table.
- Figure 2, for the quantitative data (means of the three replicates), it is better to presented them in the figure with symbols and the standard error.
- It is better also, if the authors add a table with values for the comparison between cultivars in the two phase (drought and recovery) with the p values or the level of significance for all the data analyses, for showing better the difference between cultivar in the two phases.
- The results are presented in one only paragraph. .3.1. Cultivar and sampling date interaction is presented in a single block for all variables, it is very large, it is better to divide this section into sub-sections such as, NDVI, turf cover, soil moisture,,,,, and in each sub-session compare the cultivars in the two years. it's more readable.
- The figure 3 and 4. Add the Standard deviation in the histogram, and the p values or the significance.
- Fig 3 and 4. Authors have compared some variables in the first year, and others in the second year. Why authors have not showed all results of all data in the figures. Authors must present all variables evaluated in the two years, compare and interpret.
- 2 Cultivar and N rate interaction. The article focuses on the effect of nitrogen fertilization, but this part has been very little detailed. Add more explanations of the effect (significant or not significant) on the measured variables and more references.
Finally, the topic of this manuscript is interesting; since the selection of turf species more tolerant to drought can contribute to the sustainability of water sources, but authors must improve the presentation of their results and discussion.
Author Response
The authors have presented a manuscript, which evaluated Poa pratensis cultivars under different nitrogen fertilization. Cultivars were exposed to drought stress phase followed by a recovery phase. Visual parameters, NDVI, digital images parameters and soil moisture were evaluated. The manuscript presents interesting results concerning the tolerance of Poa turf to drought, but they are some point which need to improve. Following, I have included some comments aimed to enhance the paper:
- In Material and methods line 94-95 four Kentucky bluegrass cultivars (‘Barduke’, ‘Brooklawn’, ‘Nublue Plus’, ‘Maurauder’) were seeded. Cultivars have been cited briefly. Explain why you have chosen these varieties and give some characteristics of the chosen varieties (for example, which are more resistant to water stress, the most productive…) differences between them.
Distributor companies sell cultivars with a description such as “very dark green color year-round” or “high stress resistance”. This is the reason why cultivars comparison by independent studies is needed. We do not found other studies investigating the drought stress resistance of the cultivars we tested.
The Table 1. Authors have presented levels of significance of cultivar and N application, but they have explained only the differences in the interaction. Explain all data presented in the table.
It is statistically correct to present results of the interaction when significant, not explaining the main effects. When interaction effects are significant, it means that the interpretation of the main effects is incomplete or misleading.
- Figure 2, for the quantitative data (means of the three replicates), it is better to presented them in the figure with symbols and the standard error.
Symbols have been added. We do not present standard errors to avoid multiple overlapping lines. The LSD is sufficient for accurate results interpretation.
It is better also, if the authors add a table with values for the comparison between cultivars in the two phase (drought and recovery) with the p values or the level of significance for all the data analyses, for showing better the difference between cultivar in the two phases.
A table has been added as supplementary material (Table S1).
- The results are presented in one only paragraph. .3.1. Cultivar and sampling date interaction is presented in a single block for all variables, it is very large, it is better to divide this section into sub-sections such as, NDVI, turf cover, soil moisture,,,,, and in each sub-session compare the cultivars in the two years. it's more readable.
Changed accordingly.
- The figure 3 and 4. Add the Standard deviation in the histogram, and the p values or the significance.
SE and letters have been added according to reviewer 1. The p value is reported in table 1.
- Fig 3 and 4. Authors have compared some variables in the first year, and others in the second year. Why authors have not showed all results of all data in the figures. Authors must present all variables evaluated in the two years, compare and interpret.
Results are presented for significant interaction only.
- 2 Cultivar and N rate interaction. The article focuses on the effect of nitrogen fertilization, but this part has been very little detailed. Add more explanations of the effect (significant or not significant) on the measured variables and more references.
The issue has been enlarged. However, we do not find more references, if reviewers have any specific reference in mind that could improve literature review, we’d be happy to include it.
Finally, the topic of this manuscript is interesting; since the selection of turf species more tolerant to drought can contribute to the sustainability of water sources, but authors must improve the presentation of their results and discussion.
We hope our changes according to reviewers’ s comments have improved the manuscript.
Round 2
Reviewer 1 Report
General comments
I have read the revised manuscript (agro-1223876). Entitle: Drought resistance and recovery of Kentucky bluegrass (Poa pratensis L.) cultivars under different nitrogen fertilization rates written by Cristina Pornaro et. al., for publication of agronomy. Author did the research on application of nitrogen (N) on the resistance of the different cultivars of the Kentucky bluegrass under the drought and recovery phase. In this revised manuscript author addressed all of my comment and suggestion very well. Now the manuscript is significantly improved. Author separated the research and discussion well interpretation the discussion sections. However, still the introduction sections have some lacking to flow of story. Author should to little more work in introduction section compared than before. I have some very few minor comment to further improve this manuscript specially in introduction sections.
Minor comments and suggestions
1) Line no. 29
“many different environment” this phrase is not much clear, please rephrase this. Or author may express like ‘recreational and decorative purpose’
2) Line no. 38-49
Author well describe the overall scenario of climate change and uncertainty of rainfall and variation of temperature very well. However, due to the limited of rainfall the drought stress is induce however the important climatic stress “Drought”. However, there is not the potential literature for dealing with the negative effective in line 43-44.
3) Line no. 43-43
Author should to be present the negative effect of drought stress in broad perspective because drought is the big figure of your study. Author should to be focus the text “drought reduced the morphological traits such as leaf size, root collar diameter, plant height, vegetative growth, and physiological traits such as reduction of photosynthesis and stomatal conductance” by citing the reference. “Response of drought stress in prunus sargentii and larix kaempferii seedling….https://doi.org/10.1016/j.foreco.2020.118099
4) Line no. 61-63
Author refer the articles for deal the important of the nitrogen for the plant growth after then sequentially deal the turf grass, please remember first you should deal broad perspective then visit specific crop (ie. Turfgrass) visit https://doi.org/10.1093/jxb/erx171 Journal of Experimental Botany please mention that “All plants utilize nitrogen (N) in the form of NO3- and NH4+. It is most imperative element for proper growth and development of plants which significantly increases and enhances the yield and its quality by playing a vital role in biochemical and physiological functions of plant”.
Author Response
Minor comments and suggestions
1) Line no. 29
“many different environment” this phrase is not much clear, please rephrase this. Or author may express like ‘recreational and decorative purpose’
“many different environments” have been changed with “many environments with different climate”.
2) Line no. 38-49
Author well describe the overall scenario of climate change and uncertainty of rainfall and variation of temperature very well. However, due to the limited of rainfall the drought stress is induce however the important climatic stress “Drought”. However, there is not the potential literature for dealing with the negative effective in line 43-44.
References to support the sentence “water is increasingly limited” have been added.
3) Line no. 43-43
Author should to be present the negative effect of drought stress in broad perspective because drought is the big figure of your study. Author should to be focus the text “drought reduced the morphological traits such as leaf size, root collar diameter, plant height, vegetative growth, and physiological traits such as reduction of photosynthesis and stomatal conductance” by citing the reference. “Response of drought stress in prunus sargentii and larix kaempferii seedling….https://doi.org/10.1016/j.foreco.2020.118099
Using references about trees might be out of context.
4) Line no. 61-63
Author refer the articles for deal the important of the nitrogen for the plant growth after then sequentially deal the turf grass, please remember first you should deal broad perspective then visit specific crop (ie. Turfgrass) visit https://doi.org/10.1093/jxb/erx171 Journal of Experimental Botany please mention that “All plants utilize nitrogen (N) in the form of NO3- and NH4+. It is most imperative element for proper growth and development of plants which significantly increases and enhances the yield and its quality by playing a vital role in biochemical and physiological functions of plant”.
A sentence has been added referring to the suggested reference.